# Single-cell analysis of uterosacral ligament revealed cellular heterogeneity in women with pelvic organ prolapse

Xiaochun Liu [1,4 ✉], Minna Su[1,4], Lingyun Wei[1], Jia Zhang[1], Wenzhen Wang[1], Qian Hao[2], Xiling Lin[1] & Lili Wang[3]

Pelvic organ prolapse (POP) markedly affects the quality of life of women, including significant financial burden. Using single-cell RNA sequencing, we constructed a transcriptional profile of 30,452 single cells of the uterosacral ligament in POP and control samples, which has never been constructed before. We identified 10 major cell types, including smooth muscle cells, endothelial cells, fibroblasts, neutrophils, macrophages, monocytes, mast cells, T cells, B cells, and dendritic cells. We performed subpopulation analysis and pseudo-time analysis of POP primary cells, and explored differentially expressed genes. We verified previous cell clusters of human neutrophils of uterosacral ligaments. We found a significant reduction in receptor-ligand pairs related to ECM and cell adhesion between fibroblasts and endothelial cells in POP. The transcription factors related to the extracellular matrix, development, and immunity were identified in USL. Here we provide insight into the molecular mechanisms of POP and valuable information for future research directions.

[1] Shanxi Bethune Hospital, Shanxi Academy of Medical Sciences, Tongji Shanxi Hospital, Third Hospital of Shanxi Medical University, 030032 Taiyuan, China. [2] Taiyuan Health School, 030012 Taiyuan, China. [3] Taiyuan University of Technology, 030024 Taiyuan, China. [4] These authors contributed equally: Xiaochun Liu, Minna Su. ✉email: tyxchliu@163.com

Pelvic organ prolapse (POP) is the displacement of female pelvic organs (vagina, bladder, uterus, and/or rectum) caused by the weakness of their associated supporting tissue. It is a prevalent, burdensome, and limiting disease with enormous physical and emotional discomfort and a huge economic burden[1,2]. The prevalence of POP is 3–6% worldwide[3], 19.7% in 16 developing countries[4,5], 9.67% in China[5], and 2.9% in the USA[6]. With increased longevity, it is expected that the prevalence of symptomatic POP will increase to 46% by the year 2050 in the USA[7]. Up to 29.2% of the patients who undergo prolapse surgery will undergo another surgery for genital prolapse[8]. POP is associated with age, parity, body mass index, hormonal status, diabetes, social factors, genetic predisposition, and obstetric factors[1,9,10]. However, the pathophysiology of POP remains unclear, and risk factors fail to explain the mechanism of the occurrence and development of POP without risk factors.

The uterosacral ligaments (USL) play indispensable roles in apical support of the uterus and upper vagina in POP[11,12]. Unlike the skeletal ligaments that connect the bones, USL is a visceral ligament similar to the mesenteries and is composed of smooth muscle, blood vessels, nerves, adipose tissue, and loose connective tissue[13]. Substantial evidence suggests that the pathophysiological mechanisms underlying POP are related to changes in the extracellular matrix (ECM) of supporting tissues. The USL is mainly composed of the ECM[14,15]. Histological studies of USL in patients with POP have shown that both types I and III collagen are reduced with an increased III/I ratio, and the ECM density is reduced[14,16–18]. Moreover, it has been reported that the expression of the collagen-degrading enzyme, matrix metalloproteinase (MMP), is increased, and the expression of the tissue inhibitor of metalloproteinases (TIMP) is decreased, indicating dysregulation of MMP and TIMP[14,19]. In addition, studies have shown that the increased infiltration of neutrophils in the USL is related to the occurrence of POP[20,21]. Numerous studies have reported proteomic, transcriptomic, and genomic changes in POP[14,19–24]. At the gene level, complex pathogenesis is believed to be involved in POP, including those related to changes in the ECM, immune responses, signaling pathways, and differentially expressed genes (DEGs)[25–27].

A powerful approach for high-resolution transcriptomics analysis is single-cell RNA sequencing (scRNA-seq), which provides an opportunity to decipher the heterogeneity of gene expression in individual cell populations. Although Li et al.[22] have used scRNA-seq to construct a transcriptomic atlas of anterior vaginal prolapse in POP using the vaginal wall and contributed to defining the molecular mechanism of POP, no study has analyzed the USL in POP by performing scRNA-seq thus far. Hence, this study aimed to provide a transcriptomic atlas for normal and prolapsed human USL cell types, identify DEGs in different cell types of the USL, and construct a differentiation trajectory of the primary cells. In addition, we analyzed the intercellular communication and key transcription factors (TFs) in USL. The role of USL cells in ECM remodeling and the immune response was revealed. We provided a molecular mechanism for prolapse at the single-cell level and enhanced the understanding of the pathophysiological process of POP, providing insight to improve current prevention and treatment strategies.

## Results

**Single-cell transcriptomic atlas in POP and control samples**. To dissect the cellular heterogeneity and expression characteristics of USL in patients with POP, we performed scRNA-seq of the USL in three patients with POP and one control patient who had undergone hysterectomy for hysteromyoma (Fig. 1a). After strict quality control, we obtained the transcriptomes of 30,452 single cells. Using dimension reduction and unsupervised graph clustering, we obtained 10 cell subpopulations (Fig. 1b). According to at least three well-known cell markers, including COL1A1, COL1A2, and DCN for fibroblasts, the cells were annotated into 10 cell types: smooth muscle cells (34.93%), endothelial cells (21.29%), fibroblasts (17.36%), neutrophils (4.92%), macrophages (5.36%), monocytes (4.05%), mast cells (6.60%), T cells (3.70%), B cells (0.85%), and dendritic cells (DCs) (0.94%) (Fig. 1c, Supplementary Fig. 1, Supplementary Data 1). The three most abundant cell types were smooth muscle cells, endothelial cells, and fibroblasts. The ratio of fibroblasts and smooth muscle cells in the POP group was similar to that in the control group, suggesting that abnormal gene expression may be more crucial than the ratio change. Compared to the control group, the number of macrophages, neutrophils, and mast cells in the POP group increased, and the number of endothelial cells decreased significantly, suggesting that these cells play a role in the pathophysiological mechanisms of POP (Fig. 1d, Supplementary Data 1).

To understand the functions of each cell type in the USL, we analyzed the top 10 DEGs (Fig. 1e) and performed Gene Ontology (GO) function enrichment analysis (Supplementary Fig. 2). Smooth muscle cells (SMCs) are enriched in mitochondrial pathways, whereas extracellular matrix pathways are downregulated in monocytes, mast cells, DCs, and T cells. In addition, to understand the changes in expressed genes in the POP group compared with the control group, we examined the DEGs in all cell types. Compared with the control group, the specific up- and downregulated genes of each cell type in the POP group are shown in the volcano plot (Fig. 1f). The enrichment analysis revealed that, compared with the control samples, the cell types of the POP group were enriched in the pathways related to collagen-containing ECM, muscle system processes, neutrophil activation, and other related pathways and were widely downregulated in proteins targeting the endoplasmic reticulum, ribosome biogenesis, and other pathways (Supplementary Fig. 3). Surprisingly, SMCs downregulated ECM organization pathways (Supplementary Fig. 3).

**Characterization of stromal cells**. Stromal cells are the main cells in the USL, accounting for 73.58% of the cell population. To study the subsets of stromal cells, we conducted a subpopulation analysis of three types of stromal cells (Figs. 2 and 3, Supplementary Fig. 4a). In addition, pseudo-time analysis was performed on the three types of cells.

Fibroblasts are critical cell components in the USL and play a key role in the ECM. We conducted an unsupervised analysis clustering on 5157 fibroblasts and divided them into five clusters in the heatmap (Fig. 2a), which were specifically marked as S3FRP1+ fibroblasts, FGBP2+ fibroblasts, PCP4+ myofibroblasts, PGS5+ mural cells, and SLPI+ fibroblasts (Fig. 2b). Fibroblasts3 were decreased in POP, and the ratio of other clusters did not change significantly (Fig. 2c, Supplementary Fig. 4a, Supplementary Data 1). Fibroblasts1 with the highest abundance was marked as SFRP1+ fibroblasts, which highly expressed DPT, CCDC80, COL14A1, VEGFD, FBLN1, PI16, and C7 and were identified as universal fibroblasts, which may be the same cell as DPT + PI16+ fibroblasts in the pancreas and human adipose tissue[28]. Fibroblasts2 highly expressed FGBP2, FOXD1, and CYP1B1, defined as FGBP2+ fibroblasts, and was enriched in growth factor binding and mesenchyme development pathways (Fig. 2d, Supplementary Data 1). Fibroblasts3 and fibroblasts4 highly expressed muscle contraction-related genes such as ACTA2 and ACTG2. Fibroblasts3 was marked as PCP4+

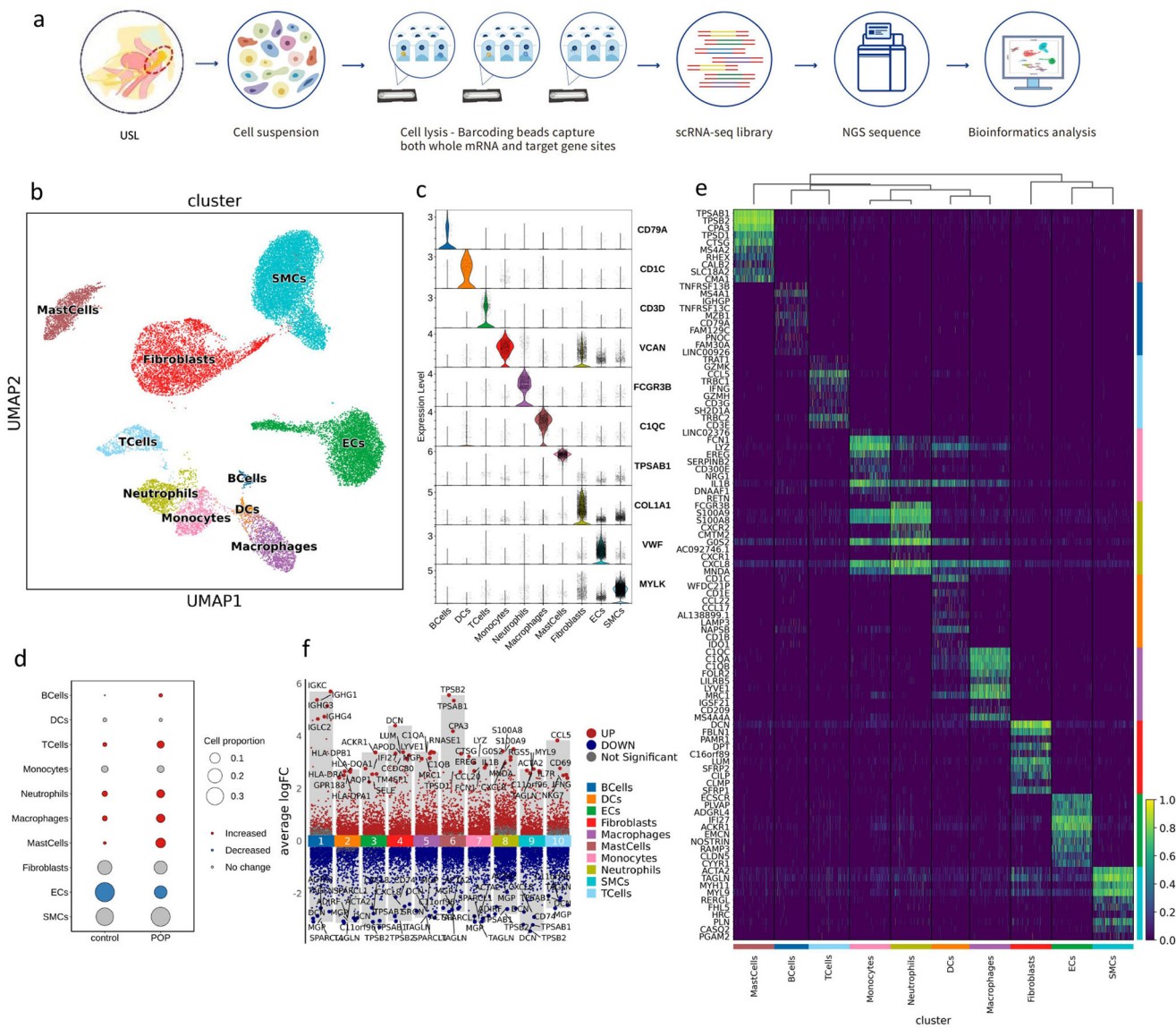

**Fig. 1 Diverse cell clusters in the uterosacral ligament with Single-cell RNA-seq analysis. a** Schematic representation showing tissue dissociation, cell suspension, cell lysis, library building, sequencing, and bioinformatics analysis. **b** UMAP plots of the 30,452 cells illustrating ten subpopulations. **c** Violin plots showing the expression of one canonical marker for each cell type. **d** Comparison of the ratio of cells in POP and control samples. **e** Heatmap showing the relative expression of the top 10 genes in each cell type. **f** Volcano plot showing the differentially expressed genes of POP compared with the control sample. SMC smooth muscle cells, EC endothelial cells, DCs dendritic cells.

myofibroblasts. Fibroblasts4 was identified as *RGS5*+ mural cells by known markers (*RGS5*, *MCAM*, *NOTCH3*, and *PDGFRB*)[29] (Supplementary Fig. 5) and was enriched in the pathway of oxidative phosphorylation and vascular process in the circulatory system. Fibroblasts5 was marked as *SLPI*+ fibroblasts, which was considered to be a fibroblast related to inflammation, with high expression of *WNT10B*, *MFAP5*, *SCARA5*, and *ClEC3B*. GO analysis showed *SLPI*+ fibroblasts not only enriched the pathways related to extracellular mechanism but also enriched neutrophil activation and neutrophil degranulation (Fig. 2d). Furthermore, we performed a pseudo-time analysis for fibroblast subtypes (Fig. 2e). *SFRP1*+ fibroblasts and *FGBP2*+ fibroblasts that regulate the ECM were widely distributed. *PCP4*+ myofibroblasts and *RGS5*+ mural cells were mainly distributed at the beginning of the track, and genes related to muscle contraction, such as *MYH11* and *ACTA2*, were highly expressed as well (Fig. 2f). SLPI+ fibroblasts were mainly distributed on the right side of the trajectory. In the heatmap of genes that varied in

expression over pseudo-time, collagen genes were upregulated at the end of the trajectory. As pseudo-time analysis does not fully reveal the developmental dynamics of differentiation, we further applied RNA velocity analysis. The *FGBP2*+ fibroblasts had RNA velocity vectors pointing towards the fibroblasts1, and then the RNA velocity vectors of *SFRP1*+ fibroblasts pointed towards the fibroblasts5 (Fig. 2g).

SMCs were the most abundant cells in the USL. A total of 10,377 SMCs were divided into four cell clusters (Fig. 3a). We found that, compared with normal tissues, the proportion of SMCs2 in POP increased significantly, whereas the expression of SMCs3 decreased (Fig. 3b-c, Supplementary Data 1). All four clusters highly expressed stress-related genes, including *ATF3*, *DNAJB1*, *HSPA1A*, *HSPA1B*, and *FOSB* (Supplementary Fig. 4b). SMCs1 is the main cluster expressing stress-related genes. According to its DEGs, SMCs1 was marked as *ATF3*+SMCs. *ATF3*, *FOS*, and *FOSB*, stress-related TFs, were identified as key TFs of *ATF3*+SMCs (Supplementary Fig. 4b). Additionally,

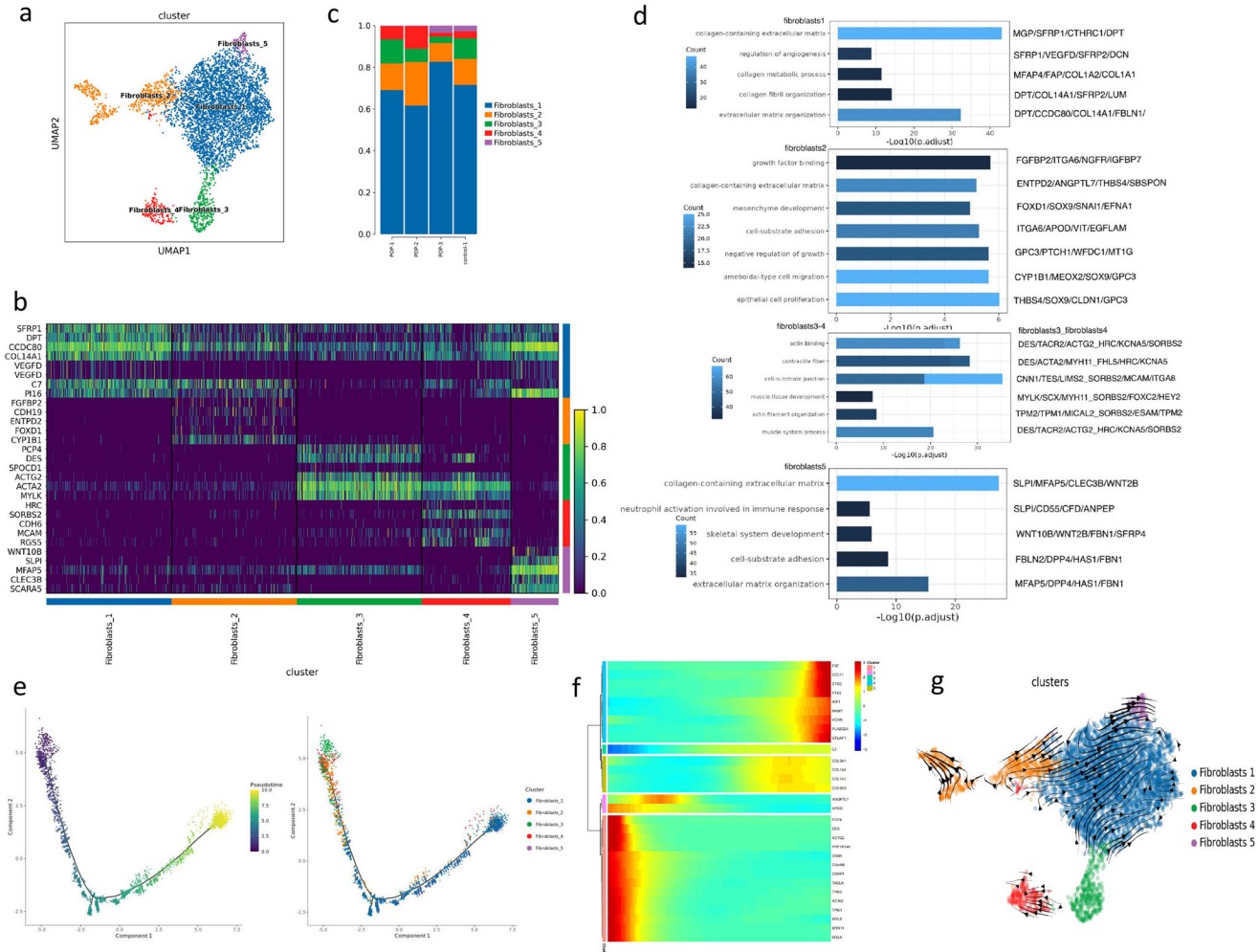

**Fig. 2 Subclustering of fibroblasts in USL reveals cellular heterogeneity. a** UMAP plot showing the subclusters of fibroblasts. **b** Heatmap showing the differentially expressed genes of five fibroblast subclusters. **c** Bar plots showing the percentage of the subclusters of fibroblasts in control and POP samples. **d** Bar plots displaying GO enrichment of upregulated genes of five fibroblast subclusters in USL samples. The color represents the *P*-value. **e** Trajectory reconstruction of all single cells of five fibroblast subclusters and the distribution of five fibroblast subclusters on the trajectory are shown. **f** Heatmap showing the dynamic changes of gene expression with pseudo-time in fibroblasts. **g** Velocities of five fibroblast subclusters shown on the UMAP plot.

*ATF3* + SMCs enriched the regulation of hematopoiesis and the fat cell differentiation pathway (Fig. 3d, Supplementary Data 1). The highly expressed DEGs of SMCs2 were not significant. SMCs2 was marked as *CTR9* + SMCs temporarily. Unfortunately, we did not get any upregulated GO pathway from *CTR9* + SMCs. SMCs3 was marked as *CFHR1* + SMCs and was enriched in the regulation of the immune effector process pathway. Combined with oxidized low-density lipoprotein, *CFHR1* can increase the production of inflammatory cytokines by monocytes and neutrophils through activation of pyrin domain-containing protein 3 (*NLRP3*)[30,31]. Furthermore, *CFHR1* + SMCs were enriched in the ECM-associated pathway and may be of the synthetic SMC. The fourth cluster had only 23 cells, which were identified as Schwann cells according to *OX2*, *PLP1*, and *CD200*. Pseudo-time analysis showed that in the developmental trajectory of SMCs, the SMCs3 cluster was mainly distributed in the early stage, whereas *ATF3* + SMCs and *CTR9* + SMCs were mainly distributed in the late stage (Fig. 3e). The RNA velocity analysis showed that the SMCs3 had RNA velocity vectors pointing towards the *CTR9* + SMCs, and then the RNA velocity vectors of *CTR9* + SMCs pointed towards the *ATF3* + SMCs (Fig. 3f).

Endothelial cells accounted for 21.29% of all the cells. Three subgroups were subdivided into lymphatic endothelial cells (LECs), arterial endothelial cells (AECs), and venous endothelial cells (VECs) (Fig. 3g). The proportion of AECs increased in the POP group (Fig. 3h–i, Supplementary Data 1). We performed GO functional pathway enrichment analysis on endothelial cells (Fig. 3j, Supplementary Data 1). AECs were enriched in the actin-binding pathway. VECs express proteins targeting the ER pathway. LECs were associated with the collagen-containing ECM and neutrophil activation involved in the immune response.

**Characterization of myeloid cells.** Differences in the type of immune cell infiltration between the POP and control groups were significant. Myeloid cells comprised the largest cell group, except for stromal cells in the USL. There was an increase of macrophages, neutrophils, and mast cells in the POP group (Fig. 1d). Four clusters were obtained by unsupervised clustering analysis of macrophages (Fig. 4a), which all highly expressed stress response genes, such as *DNAJB1*, *HSPA1A*, *HSPA1B*, and several cytokine genes, including *CXCL2* and *CXCL8* (Supplementary Fig. 4c). Proportionally, macrophages1, 2 and 4 were strongly increased in POP samples, and macrophages3 were significantly decreased (Fig. 4b, c, Supplementary Data 1). According to several simple markers, the macrophages in USL cannot be well classified into proinflammatory M1 and anti-

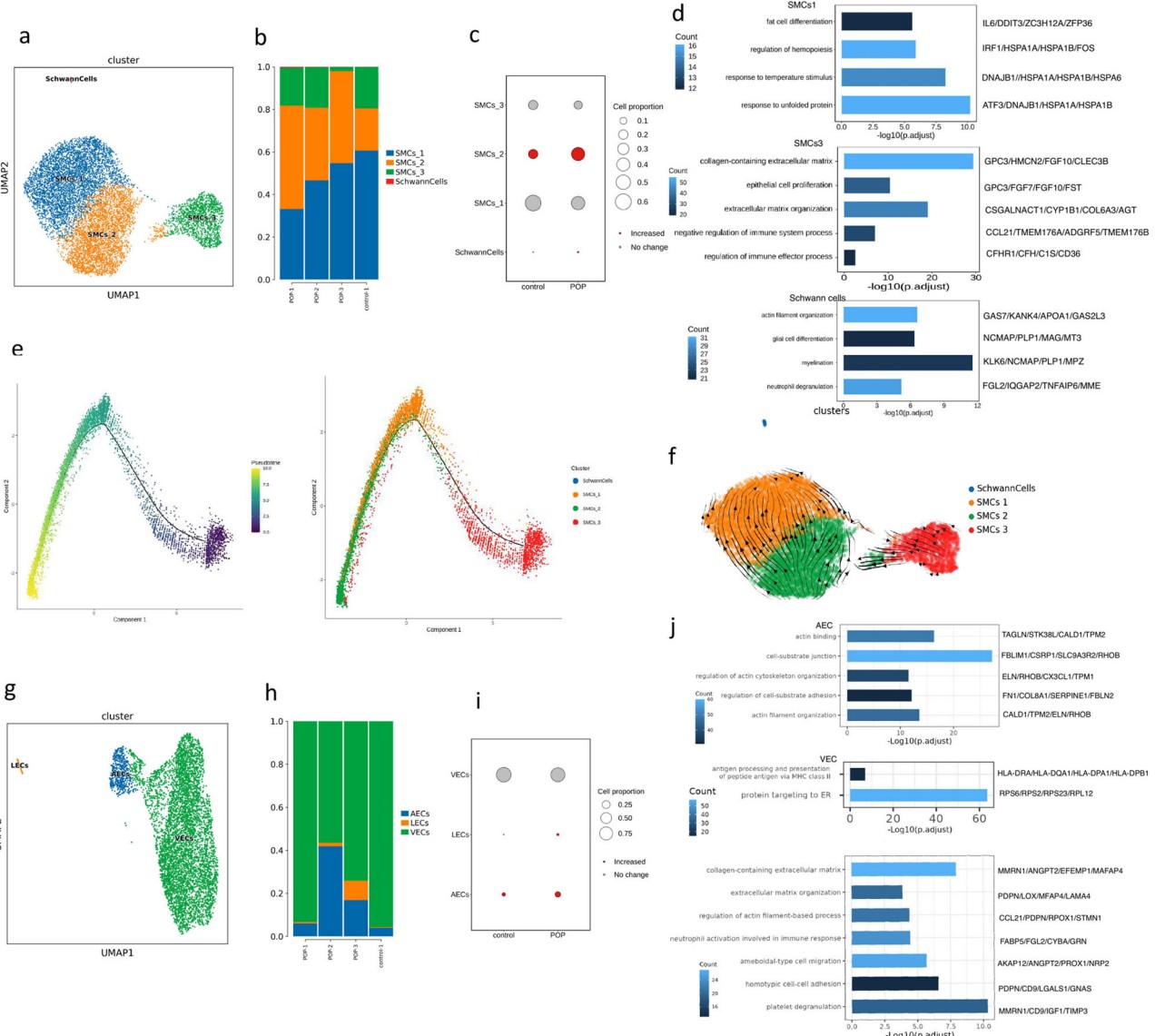

**Fig. 3 Subclustering of SMCs and ECs in USL reveals cellular heterogeneity. a** UMAP plot showing the subclusters of SMCs. **b** Bar plots showing the percentage of the subclusters of SMCs in control and POP samples. **c** Dot plot displaying relative changes in cell ratios of the subclusters of SMCs across control and POP samples. **d** Bar plots displaying GO enrichment of upregulated genes of SMC subclusters in USL samples. **e** Trajectory reconstruction of all single cells of four SMC subclusters and the distribution of four SMC subclusters on the trajectory are shown. **f** Velocities of four SMC subclusters are shown on the UMAP plot. **g** UMAP plots showing the subclusters of ECs. **h** Bar plots showing the percentage of the subclusters of ECs in control and POP samples. **i** Dot plot displaying relative changes in cell ratios of the subclusters of ECs across control and POP samples. **j** Bar plots displaying GO enrichment of upregulated genes of EC subclusters in USL samples.

inflammatory M2 categories[32]. M1marker *IL1B* is mainly distributed in macrophages3 and expressed as a part of *TNF* and *NFKB1* (Supplementary Fig. 5);[33,34] hence, it is considered that macrophages3 may be a M1 subtype. Pathways such as neutrophil activation and leukocyte chemotaxis were enriched in macrophages3 (Fig. 4d). Macrophages1 and macrophages2 have similar characteristics, both of which express a large number of stress-related genes, and neither can be clearly distinguished as M1 or M2 subtype. M1marker *CD68*, M2marker *MRC1*, *CD163*, *TGFB1*, *MERTK*, and *STAB1* (Supplementary Fig. 5)[35] were distributed in macrophages1 and macrophages2; however, they were more inclined to anti-inflammatory M2 subtype. GO function analysis showed that macrophages1 enriched stress-related pathways, and macrophages2 enriched lipoprotein particle binding and cell-matrix adhesion pathways (Fig. 4d, Supplementary Data 1). The

fourth cluster had only 22 cells and highly expressed *CHIT1*, a marker of macrophage activation related to fibrosis[36]. Monocytes were divided into four clusters (Fig. 4e). Monocytes1 were significantly increased in POP samples and monocytes3, and 4 were significantly decreased in POP samples (Fig. 4f-g, Supplementary Data 1). We found almost all monocytes expressed *CD14*, and only monocytes3 expressed *CD16/FCGR3A* (Supplementary Fig. 5). According to this, we suggested monocytes3 were intermediate monocytes (*CD14+ CD16+*) and the rest were classical monocytes (*CD14+ CD16−/low*). Intermediate monocytes have different transcription characteristics from other monocytes. Monocytes3 was enriched in the pathway oxidative phosphorylation, RNA catabolic process, and neutrophil activation (Fig. 4h, Supplementary Data 1), which is consistent with the research of Qian et al.[37]. Monocytes1 highly expressed inflammation-related

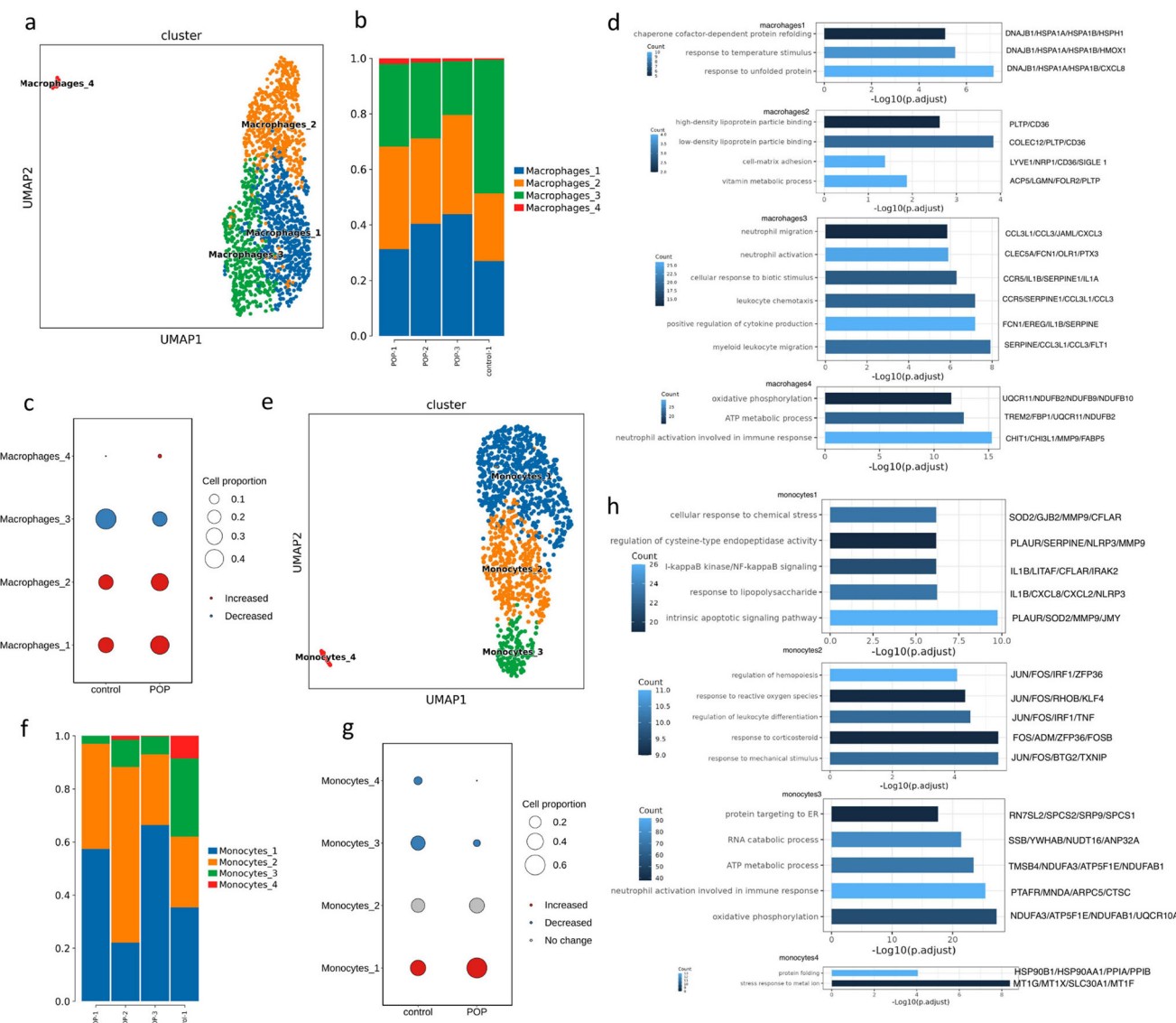

**Fig. 4 Subclustering of macrophages and monocytes in USL reveals cellular heterogeneity. a** UMAP plot showing the subclusters of macrophages. **b** Bar plot showing the percentage of four macrophage subclusters. **c** Dot plot displaying relative changes in cell ratios of the subclusters of macrophages across control and POP samples. **d** Bar plots displaying GO enrichment of upregulated genes of four macrophages subclusters in USL samples. **e** UMAP plot showing the subclusters of monocytes. **f** Bar plot showing the percentage of four monocytes subclusters. **g** Dot plot displaying relative changes in cell ratios of the subclusters of monocytes across control and POP samples. **h** Bar plots displaying GO enrichment of upregulated genes of four monocytes subclusters in USL samples.

genes such as *IRAK2*, *CD72*, and *CCL20* and was enriched in I-kappaB kinase/NF-kappaB signaling, intrinsic apoptotic signaling pathway, and so on (Fig. 4h, Supplementary Fig. 4d). Monocytes2 highly expressed *JUN*, *FOS*, and *ATF3* transcription factors and early response gene family (*EGR1*, *IER2*), which were enriched in stress-related pathways. Monocytes4 was mainly enriched in the reaction of metal ions. DCs highly expressed major histocompatibility complex-related genes, which is consistent with the characteristics of DCs as antigen-presenting cells, and were divided into cDC1, cDC2, and matureDC3 (Supplementary Fig. 6a).

In addition, we performed a pseudo-time analysis of the 11 cell clusters of mononuclear phagocytes, and there were 2 branching points in the trajectory (Fig. 5a, b). The classical monocytes were mainly located in states 1, 2, and 3; intermediate monocytes were located in states 2, 3, and 5; macrophages3 that were possibly an M1 subtype were mainly located in states 4 and 5; and the

remaining three macrophage clusters were mainly located in state 4; DCs were located in state 2 and state 5. Pseudo-time analysis showed the development track from classical monocytes to intermediate monocytes, then to macrophages or DC. We further identified genes that varied in expression over pseudo-time. *IL1RN*, *IL1B*, and *CCL20* were mainly upregulated at the beginning of the pseudo-time, and *FOS*, *JUN*, and complement-related genes (*C1QA*, *C1QB*, and *C1QC*) were upregulated near the end (Fig. 5c). We further conducted the RNA velocity analysis, and found that both monocytes and macrophage had RNA velocity vectors pointing towards the DCs (Fig. 5d).

Further analysis of 1136 neutrophils in the POP group revealed four clusters with different gene expression profiles (Fig. 5e). The number of neutrophils in the POP group increased (Fig. 5f, Supplementary Data 1), consistent with the results of previous studies[20,21]. Xie et al. mapped neutrophil subpopulations G0–G5 to progressively maturing neutrophils in bone marrow, tissues,

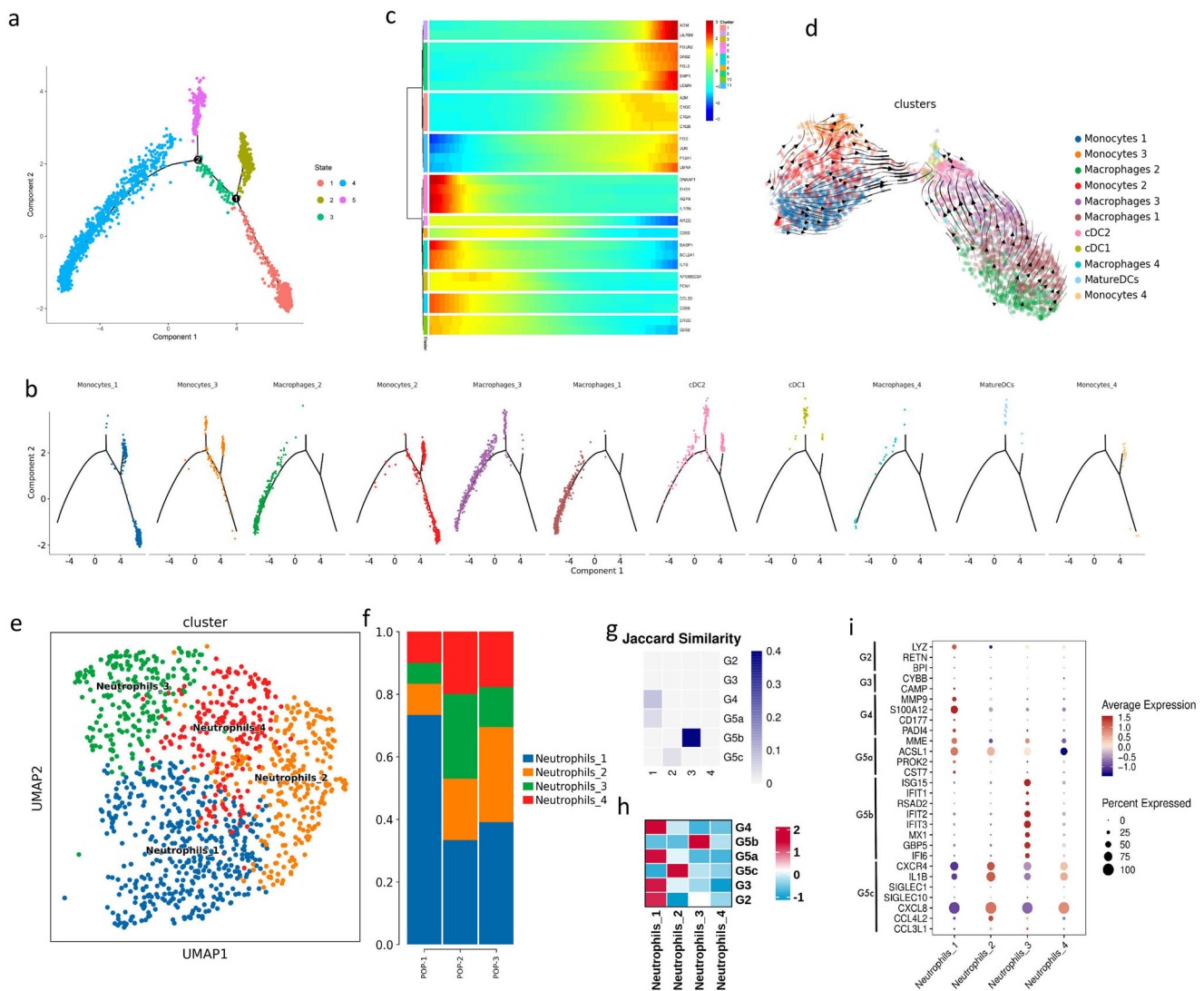

**Fig. 5 The pseudo-time analysis of mononuclear phagocytes and the heterogeneity of neutrophils. a** The state of pseudo-time trajectory of mononuclear phagocytes. **b** The distribution of eleven subclusters of mononuclear phagocytes on the trajectory is shown. **c** Heatmap showing the dynamic changes of gene expression with pseudo-time in mononuclear phagocytes. **d** Velocities of mononuclear phagocytes shown on the UMAP plot. **e** UMAP plot showing the subclusters of neutrophils. **f** Bar plot showing the percentage of four neutrophils subclusters. **g** Heatmap showing correlation of scRNA-seq defined neutrophil clusters with the neutrophil subtypes reported by Xie et al. **h** Heatmap showing the gene set analysis of our scRNA-seq defined neutrophil clusters DEGs with the DEGs reported by Xie et al. **i** Dot plot showing the DEGs expression of neutrophil clusters reported by Chen et al. in our scRNA-seq defined neutrophil clusters.

and circulation[38]. Using the DEGs provided by Xie et al., we conducted correlation analysis and gene set analysis (Fig. 5g, h), and found that Neutrophils1 had the characteristics of G2-5a, Neutrophils2 were in accord with G5c, and neutrophils3 were in accord with G5b subpopulations. Correlation analysis showed Neutrophils4 were different from any subpopulation of G0–G5. However, according to the known neutrophil typing markers provided by Chen et al.[39], we suggested that Neutrophils4 had the characteristics of G5c (Fig. 5i, Supplementary Data 1). Neutrophils1 had the characteristics of G3-5a which were relatively immature neutrophils compared with G5b and G5c, and highly expressed gelatinase granule gene MMP9, and enriching neutrophil degranulation/neutrophil activation pathway. Neutrophils3 expressed a large number of interferon-induced proteins (*RSAD2*, *IFIT1*, *IFIT3*, *HERC5*, *MX1*) and were enriched in interferon-related pathways (Supplementary Fig. 4e). These characteristics were consistent with G5b in tissues and umbilical cord blood. Neutrophils2 secreted a large number of cytokines,

such as *IL1B*, *CCL4L2*, *CCL4*, *IL1A*, *IL1RN*, *CXCL1*, and *CXCL2*, and enriched in response to lipopolysaccharide and cellular response to interleukin-1 pathway. Neutrophils4 only has 47 DEGs, indicating the similarity between neutrophils4 and other subpopulations.

Mast cells highly expressed *TPSAB1*, *TPSB2*, and *CPA3*. GO analysis showed that the upregulated pathways in mast cells were related to immune cell activation, while significantly down-regulated ECM and other pathways (Supplementary Fig. 2). The number of lymphocytes was relatively small, and the number of T cells was significantly higher in the POP group than in the control group (Fig. 1d). Cluster analysis of T cells revealed naïve T cells, CD8 effector T cells, and NKT cells (Supplementary Fig. 6b).

**Cell–cell interaction patterns in POP and control samples.** We performed cell–cell interaction analysis using CellphoneDB in the USL. In all groups, intensive interaction between fibroblasts and

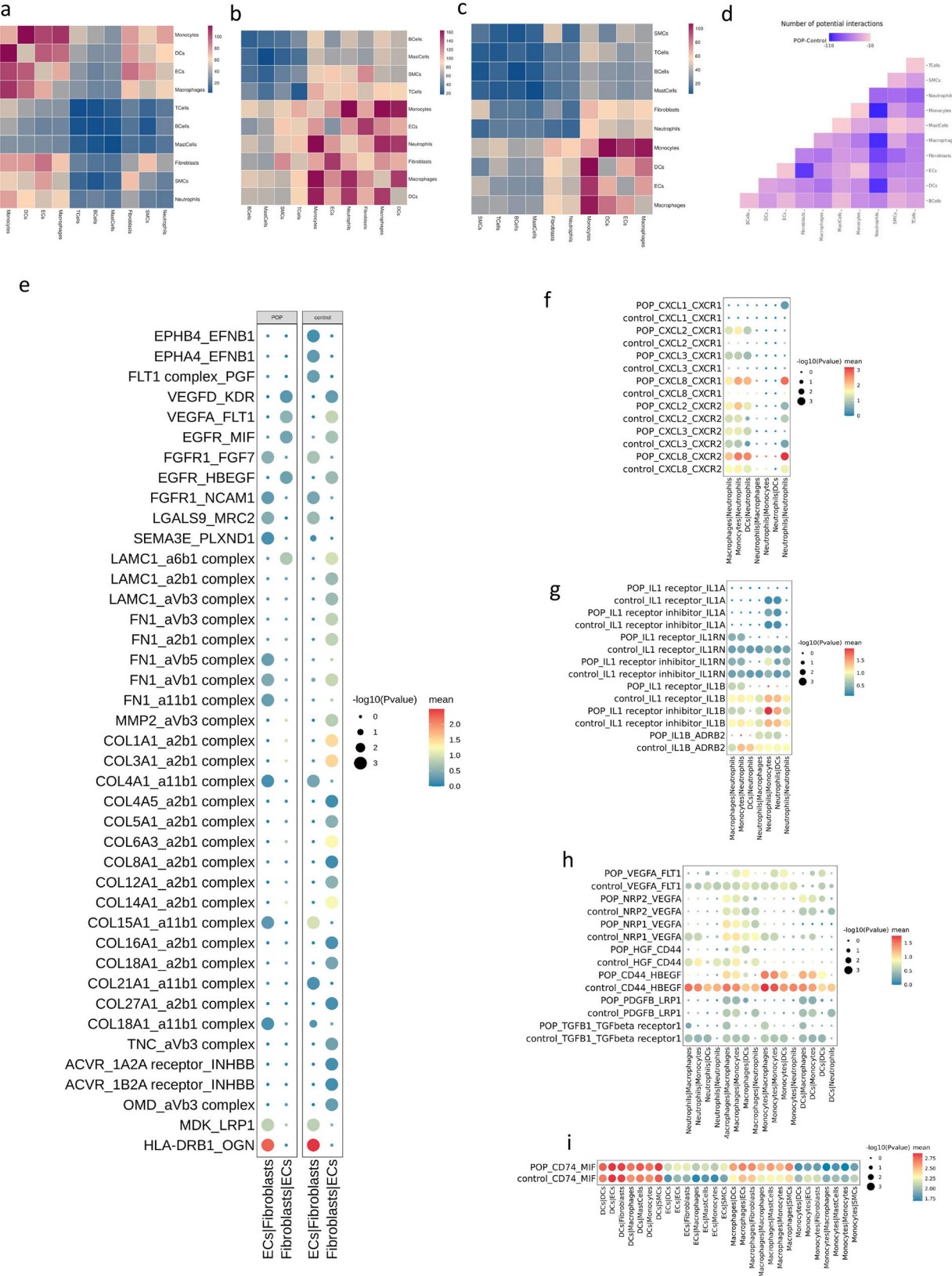

endothelial cells was observed in stromal cells (Fig. 6a). In immune cells, we observed a dense communication network among neutrophils, macrophages, DCs, and monocytes, suggesting the importance of these cell–cell interactions in the USL (Fig. 6a). In addition, we identified receptor–ligand pairs in the control and POP groups respectively (Fig. 6b, c). Cell–cell interactions in the POP group were attenuated compared to the control group, especially between fibroblasts and ECs, neutrophils, and MPs (Fig. 6d).

We assessed the specific interactions between ECs and fibroblasts (Fig. 6e, Supplementary Data 2). In the control group, fibroblasts secreted a large number of collagen-like genes such as

**Fig. 6 Potential ligand–receptor interactions analysis in USL. a** Heatmap showing the numbers of cell–cell interacting pairs with each other in USL. **b** Heatmap showing the numbers of cell–cell interacting pairs with each other in control group. **c** Heatmap showing the numbers of cell–cell interacting pairs with each other in the POP group. **d** Heatmap depicts the changed numbers of cell–cell communications in POP samples compared with control samples (all decreased). **e** Dot plots depicting ligand–receptor pairs between fibroblasts and ECs in control sample and POP samples. Dot size indicates the significance of the interaction and the spectrum of color represents the intensity of the interaction. **f** Chemokines interacting pairs between neutrophils and mononuclear phagocytes in POP and control group. **g** IL-1 family interacting pairs between neutrophils and mononuclear phagocytes in the POP and control group. **h** Growth factors interacting pairs among neutrophils and mononuclear phagocytes in POP and control group. **i** CD74-MIF ligand–receptor pairs in the POP and control group.

*MMP2*, *FN1*, *LAMC1*, and *OMD* to interact with endothelial cells. In addition, *EPHB4_EFNB1*, *FLT1 complex_PGF*, *VEGFA_FLT1*, and *VEGFD_KDR* receptor–ligand pairs promoted endothelial cell growth. *FLT1* and *KDR* encode members of the vascular endothelial growth factor receptor (*VEGFR*) family and play critical roles in angiogenesis and vasculogenesis[40]. ECs and fibroblasts promote fibroblast growth and development through *FGFR1_FGF7* and *FGFR1_NCAM1* receptor–ligand pairs. Above, we reported that the proportion of ECs was reduced in the POP group compared with that in the control group. Accordingly, the interaction between ECs and fibroblasts was further significantly reduced in the POP group (Fig. 6d), which was mainly reflected by the significant reduction in receptor–ligand pairs related to ECM and cell adhesion. However, there was no significant difference in *VEGF* or *FGF* levels, and a large number of growth factor interaction pairs were still enriched in the POP group.

Compared with the control group, neutrophils increased significantly in the USL of the POP group (Fig. 1d), and there was a rich cellular communication network between neutrophils and mononuclear phagocytes. We found that, in the POP group, mononuclear phagocytes secreted large amounts of *CXCL2*, *CXCL3*, and *CXCL8*, which were bound to *CXCR1* in neutrophils, thus attracting neutrophils to the USL site (Fig. 6f, Supplementary Data 2). Several interleukin-1 (*IL-1*) family gene interactions were detected between neutrophils and other immune cells. We found that the interaction of *IL1B_ADRB2* in the POP group was less than that in the control group (Fig. 6g, Supplementary Data 2). These four cell types regulate the ECM through *HGF_CD44* and *CD44_HBEGF* receptor-ligand pairs and then regulate cell adhesion and migration (Fig. 6h, Supplementary Data 2). The *CD74_MIF* receptor–ligand pair was highly expressed in immune cells other than neutrophils. The interaction of *CD74_MIF* in the POP group was slightly higher than that in the control group (Fig. 6i, Supplementary Data 2). *MIF-CD74* can activate the mitogen-activated protein kinase (MAPK) signal transduction pathway, cause inflammatory responses, and thus regulate the inflammatory microenvironment of damaged tissue. In addition, we found several growth factors that were highly expressed in USL (Fig. 6h). Among them, *TGFB1* had been proven to attenuate the loss of ECM by inhibiting the activities of *MMP-2/9* through the *TGF-β1/Smad3* signaling pathway in USL and was inversely correlated with the severity of POP[41].

**Transcriptional regulation in POP and control samples.** Combinatorial interactions between TFs lead to tissue-specific gene expression, which determines cell identity and maintains cell homeostasis. We used single cell transcription factor regulatory network analysis (SCENIC) to assess the expression of TFs in the USL and to identify key TFs. First, we investigated the expression of TFs and cell-type-specific TFs in the control group. Highly expressed NFATC2 in macrophages, neutrophils, and monocytes, GATA3 in mast cells and T cells, and STAT4 in T cells can all promote T-cell differentiation (Fig. 7a, Supplementary Fig. 7). High basal expression of STAT4 contributes to the NKT cell responses[42]. GO functional enrichment analysis was performed

on these TFs, and MYBL2, GATA3, and ARID3A were enriched in lymphocyte differentiation and regulation of myeloid leukocyte differentiation (Fig. 7b, Supplementary Data 2).

Furthermore, we investigated the TFs in the POP group. The specific TFs in neutrophils were the same as those in the control group; however, the expression intensity was significantly enhanced (Fig. 7a, b, Supplementary Fig. 7). In addition to neutrophils, we observed that except for the TFs already highly expressed in the control group, another group of TFs was specifically highly expressed in the POP group. Several specific TFs of monocytes, such as OLIG1, CEBPE, POU2F2, and NFE2, were all enriched in neutrophil activation (Fig. 7c-d, Supplementary Data 2). C/EBP family is related to neutrophil proliferation, differentiation, maturation, and granular protein formation[43]. CEBPE and NFE2 were specific TFs of neutrophils. CEBPE could promote the terminal differentiation of neutrophils[44]. NFE2L3 was specifically active in mast cells and was enriched in ECM organization, suggesting that the increase in mast cells in POP samples is critical for the development of ECM remodeling and prolapse.

In addition, TFs of stromal cells in the control and POP groups tended to be functionally similar, and both were related to the development, differentiation, and proliferation of specific cells. However, the TF function of the stromal cells in the POP group was more closely related to cell function. Fibroblast-specific TFs, such as PBX1, ARRM3, and AR, were significantly upregulated, all of which are related to ECM remodeling. The SMC-specific TFs FOXC2, TBX2, EMX2, and ISL1 play important roles in muscle development. ZNF233, SP6, and SMAD1 were highly active in endothelial cells and contribute to vascular development t (Fig. 7). These potential upstream regulators contribute to our understanding of the pathogenesis of POP.

## Discussion

POP markedly affects the health and quality of life of women, and there is no optimal treatment. The uterosacral ligament, which provides apical support for the upper part of the uterus and vagina, plays an integral role in POP. Although POP has been extensively studied, its key mechanisms have not been fully elucidated; in particular, the role of immune cells, including macrophages, monocytes, DCs, and neutrophils, in POP pathology is unclear. To the best of our knowledge, we performed a scRNA-seq analysis using samples from the USL in POP patients and control patients for the first time to elucidate the cellular composition of the USL and the expression of specific genes for different cell types and to provide a better understanding of the pathophysiological mechanisms and the immune microenvironment of POP. We identified 10 cell types, including 3 stromal and 7 immune cells. We performed unsupervised graph clustering, GO functional enrichment analysis on these cells, and pseudo-time analysis of some cells, which revealed heterogeneity between the main mesenchymal cells and immune cells. In addition, we found alterations in cell–cell interactions and TFs in POP. Overall, our study greatly deepens our understanding of the

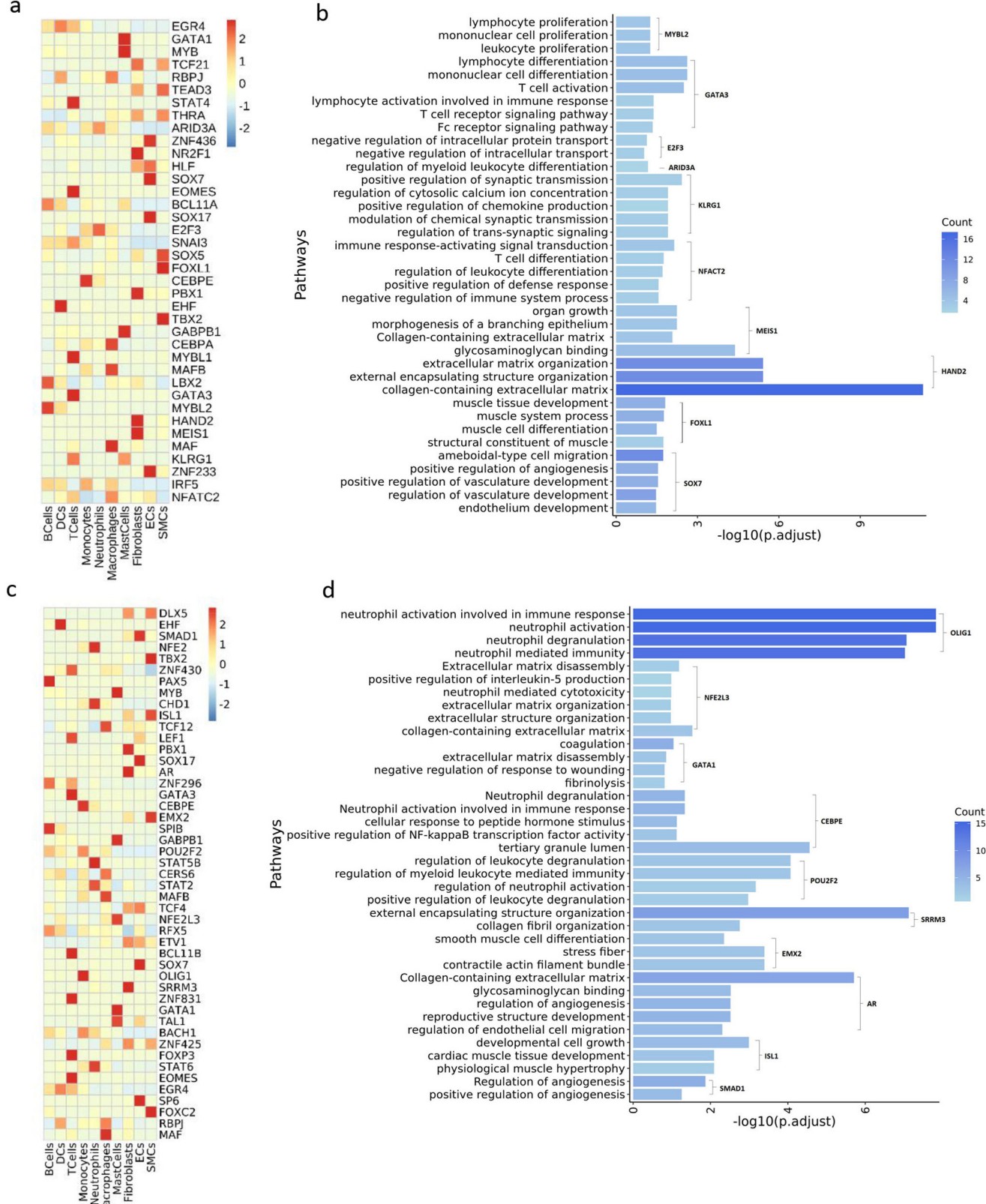

**Fig. 7 Cell type-specific TFs and GO analysis of representative TFs. a** Heatmap of representative upregulated TFs in the control samples. **b** Bar plots indicating the GO enrichment results of representative TFs and corresponding target genes in control samples. **c** Heatmap of representative upregulated TFs in the POP samples. **d** Bar plots indicating the GO enrichment results of representative TFs and corresponding target genes in POP samples.

pathological mechanisms of POP and provides a theoretical basis for the development of related therapeutic strategies.

Previous studies have suggested that the main component of USL is the ECM[13–15]; however, we still found a large number of cells in USL. Fibroblasts, SMC, and endothelial were the main cell types of USL, accounting for 17.36%, 34.93%, and 21.29%, respectively. We verified the universal phenotype and differentiation of fibroblasts proposed by Buechler et al.[28]. DEGs analysis showed that *SFRP1*+fibroblasts, which were widely present in the USL, highly expressed the upregulated genes such as *DPT*, *C7*, *MFAP4*, and *LUM*, of universal fibroblasts. The trajectory of fibroblast subtypes indicated that universal fibroblasts differentiated into specialized fibroblasts (*SLPI*+ fibroblasts) during development. In addition to enriched functions such as the organization of the ECM and muscle system process, clustering analysis revealed that the subgroups SLPI+ fibroblasts, ATF3+ SMCs, and LEC were enriched in neutrophil activation, immune response, and neutrophil degranulation, respectively. Immune cells differed markedly between the POP and control samples. Neutrophils, macrophages, mast cells, and T cells increased significantly in the POP group. Based on several clusters enriched in pathway neutrophils activation, we did find that neutrophils increased significantly at the same time. We divided neutrophils into four clusters and found that these four clusters can correspond to the neutrophils in Chen and Xie et al.'s scRNA-seq studies[38,39], which further verified the phenotype of neutrophils in the human body. In addition, according to a widely used previous classification scheme, macrophages were grouped into pro-inflammatory (M1) and anti-inflammatory (M2) phenotypes. However, several markers cannot separate macrophages well. Some macrophage clusters expressed both M1 and M2 markers. According to the distribution of M1 and M2 markers, we found that macrophages3 had the characteristics of M1 which were decreased in POP samples, and the remaining cell population were more inclined to M2 phenotype; however, it cannot be strictly considered as M1/M2 phenotype.

In addition, we found that smooth muscle cells and macrophages in USL showed strong expression of stress-response genes[45]. SMCs3-specific marker *CFHR1* can bind to necrotic oxidized low-density lipoprotein and strongly induces inflammasome *NLRP3* in monocytes and neutrophils by exposure to *EMR2* and then secretes pro-inflammatory cytokines such as *IL-1β*, *IL-6*, *IL18*, and *TNFα*[30,31]. In this study, the expression of *EMR2/ADGRE2* was observed in macrophages, and the high expression of *NLRP3* was observed in monocytes1 and neutrophils 2. Furthermore, cell–cell interaction analysis found that many *IL-1* interactions happened between neutrophils with other cells. This may be a point in the pathogenesis of POP stress.

After sequencing the USL samples, we identified a high percentage of endothelial cells in all USL samples. Compared with the control sample, there was a significantly lower percentage of endothelial cells in the POP samples. At the same time, endothelial cells and fibroblasts showed active cell–cell interactions in the control group but were less active in the POP group. This change was reflected mainly by the high expression of ECM-related interaction pairs between fibroblasts and ECs in the control group, which was significantly reduced in the POP group. In addition, we predicted that there was a symbiotic relationship between fibroblasts and ECs in a two-cell circuit in both the control and POP groups and that they interacted mutually to obtain growth factor signals. Endothelial cells promote fibroblast development by providing *FGFR1*, whereas fibroblasts promote proliferation, migration, and differentiation of ECs by supplying *VEGF* and *VEGFD*. Furthermore, monocytes and macrophages recruit neutrophils through the interaction of *CXCL3* and *CXCL8* with *CXCR1* and *CXCR2*. Neutrophils express many *IL-1* family

genes that have been shown to play a key role in inflammation, fibrosis, and autoimmune reactions[46,47]. Therefore, *IL-1* family interaction may be a reason for promoting collagen deposition of the USL in the POP. In addition, we found that *CD74_MIF* receptor-ligand pairs are widely expressed in immune cell interactions and may activate the expression of factors related to proliferation and fibrosis of fibroblasts, such as EGFR and TGFB1[48–50]. That is, *MIF-CD74* may be a crucial receptor–ligand pair linking immune and extracellular remodeling mechanisms in POP. In conclusion, we believe that the decreased interaction between fibroblasts and ECs, neutrophils, and MPs may play an important pathogenesis of POP, and have identified several critical receptor–ligand pairs for fibroblast–immune cell interactions. Solé-Boldo et al.[51] conducted scRNA-seq of skin samples from young and aging people and found that aging led to a significant decrease in cell-to-cell interaction. Our data show that the cell communication in the POP group is less than that in the control group, especially between fibroblasts and ECs. We suspect it is aging that caused this change and aging is the key mechanism of POP.

TFs activate or repress gene transcription by recruiting other TFs, coactivators, or corepressors to affect the cell state. We found that *FOS* and *JUN* were significantly overexpressed in monocytes2 and mast cells and that *JUNB* was highly expressed in macrophages1 and monocytes2. These TFs may increase the inflammatory responses of macrophages, monocytes, and mast cells. Furthermore, we identified a series of TFs related to cell development and proliferation, including *EOMES*, *NFE2*, *PBX1*, *SOX7*, and *TBX2*. In addition, we found that TFs related to neutrophil activation were specifically active on monocytes in POP samples. Moreover, GO functional pathway analysis of DEGs in monocytes was enriched in the neutrophil activation-related pathways, indicating that monocytes may be a key cell type regulating neutrophil activity. Furthermore, we found a critical upstream TF, *NFE2L3*, which was enriched in ECM-related pathways in mast cells. Simultaneously, GO analysis showed that mast cells were significantly downregulated in pathways such as ECM organization, indicating that mast cells may play crucial roles in the regulation of ECM in the USL. Collectively, we propose that the ECM interacts with the immune system, possibly causing the accumulation and remodeling of the ECM, and the accumulation of ECM leads to the proliferation of immune cells, leading to a sustained inflammatory response[52,53].

In this study, we integrated scRNA-seq datasets from three POP samples and one hysteromyoma control sample. Since the USL tissue from resected uterine specimens in patients who underwent total hysterectomy for benign reasons is not large enough, we had insufficient samples for the control group. The sample size of the control group was smaller than that of the POP group, which may have limited the statistical power of this comparison. Furthermore, the ethnic and racial matching of the POP and control groups was poor. All patients were of Han nationality. Additional control specimens from different sources should be evaluated in future studies to address these limitations. Furthermore, because adipocytes are very light, scRNA-seq technology cannot obtain information on adipocytes, which may lead to the loss of adipocyte composition and heterogeneity in the USL. Single-cell nuclear RNA sequencing technology is required to provide further elucidation. Finally, this study lacked verification tests to prove the relevant mechanisms mentioned in this paper. More verification tests are needed to prove potential pathogenesis in the future.

To the best of our knowledge, our study was the first to explore the cell type composition and heterogeneity of POP and normal USLs by using scRNA-seq, revealing a picture of immune cell infiltration in USL tissue of POP samples. We had verified some

previous cell clusters of human fibroblasts and neutrophils. In addition, we found that fibroblasts and endothelial cells promoted the growth, proliferation, and development of each other and described how immune cells and fibroblasts communicate with each other through receptor-ligand pairs. Furthermore, we identified important TFs that regulate the ECM, cell development, and immune response. Overall, our study contributes to improving our understanding of the pathogenesis of POP and provides relevant information for future research directions and disease prevention strategies.

## Methods

**Sample collection and processing**. This study was approved by the Ethics Committee of Shanxi Bethune Hospital (YXLL-2022-131). All participants provided written informed consent to provide samples. All ethical regulations relevant to human research participants were followed. This study included women who underwent hysterectomy for POP and hysteromyoma. Patients with Pelvic Organ Prolapse Quantification (POP-Q) stage III–IV were enrolled in the study following the diagnosis by the same experienced specialist. Women who underwent hysterectomy for hysteromyoma and did not have prolapse were included as the control group. Exclusion criteria were previous pelvic reconstruction surgery, chronic pelvic inflammation, autoimmune diseases, connective tissue diseases, and malignant tumors. All patients were non-smokers, aged >50 years, and had a BMI ranging $18.5–28\,kg/m^2$. Approximately $1–1.5\,cm^3$ of the cervical attachment of the USL was collected from the hysterectomy specimen of the patient and was used for the preparation of the single-cell suspension in the next step. Clinical characteristics of POP patients and control samples were shown in Supplementary Data 1.

**Tissue dissociation and preparation**. The fresh tissues were stored in the sCelLiveTM Tissue Preservation Solution (Singleron Bio Com, Nanjing, China) on ice after the surgery within 30 min. The specimens were washed with Hanks Balanced Salt Solution (HBSS) 3 times and then digested with 2 ml sCelLiveTM Tissue Dissociation Solution (Singleron) by Singleron PythoN™ Automated Tissue Dissociation System (Singleron) at 37 °C for 15 min. Afterward, the GEXSCOPE® red blood cell lysis buffer (Singleron, 2 ml) was added, and cells were incubated at 25 °C for another 10 mins to remove red blood cells. The solution was then centrifuged at $500 \times g$ for 5 min and suspended softly with PBS. Finally, the samples were stained with trypan blue (Sigma, USA) and the cellular viability was evaluated microscopically.

**Library preparation and scRNA-seq**. Single-cell suspensions ($1 \times 10^5$ cells/ml) with PBS (HyClone) were loaded into microfluidic devices using the Singleron Matrix® Single Cell Processing System (Singleron). Subsequently, the scRNA-seq libraries were constructed according to the protocol of the GEXSCOPE® Single Cell RNA Library Kits (Singleron)[54]. Individual libraries were diluted to 4 nM and pooled for sequencing. At last, pools were sequenced on Illumina novaseq6000 with 150 bp paired-end reads.

**Quality control, dimension-reduction and clustering (Scanpy)**. Scanpy v1.8.2 was used for quality control, dimensionality reduction, and clustering under Python 3.7[55]. For each sample dataset, we filtered the expression matrix by the following criteria: (1) cells with a gene count less than 200 or with a top 2% gene count were excluded; (2) cells with a top 2% UMI count were excluded; (3) cells with mitochondrial content >30% were excluded; 4) genes expressed in less than 5 cells were excluded.

After filtering, 30452 cells were retained for the downstream analyses, with an average of 1424 genes and 4562 UMIs per cell. The raw count matrix was normalized by total counts per cell and logarithmically transformed into normalized data matrix. Top 2000 variable genes were selected by setting flavor = 'seurat'. Batch effect between samples was removed by Harmony. Cells were separated into 23 clusters by using the Louvain algorithm and setting the resolution parameter at 1.2. Cell clusters were visualized by using Uniform Manifold Approximation and Projection (UMAP).

**scRNA-seq quantifications and statistical analysis**. Raw reads were processed to generate gene expression profiles using an internal pipeline. Briefly, cell barcode and UMI were extracted after filtering read one without poly T tails. Adapters and poly-A tails were trimmed (fastp V1) before aligning read two to GRCh38 with ensemble version 92 gene annotation (fastp 2.5.3a and featureCounts 1.6.2). Reads with the same cell barcode, UMI, and gene were grouped together to calculate the number of UMIs of genes in each cell. The UMI count tables of each cellular barcode were employed for further analysis.

**Differentially expressed genes analysis**. Genes expressed in more than 10% of the cells in a cluster and with an average log (Fold Change) of >0.25 were selected as DEGs by Seurat v3.1.2 FindMarkers based on Wilcox likelihood-ratio test with default parameters.

**Cell type annotation**. The cell type identity of each cluster was determined with the expression of canonical markers found in the DEGs using SynEcoSys database. Heatmaps/dot plots/violin plots displaying the expression of markers used to identify each cell type were generated by Scanpy v1.8.2. Cell doublets were estimated based on the expression pattern of canonical cell markers. Any clusters enriched with multiple cell type-specific markers were excluded for downstream analysis.

**Pathway enrichment analysis**. To investigate the potential functions of all cell types, Gene Ontology (GO) analysis was used with the "clusterProfiler" R package 3.16.1[56]. Pathways with p_adj value <0.05 were considered significantly enriched. Gene Ontology gene sets including molecular function (MF), biological process (BP), and cellular component (CC) categories were used as reference.

**Trajectory analysis**. To map the differentiation/conversion of cell subtypes in fibroblasts, SMC, EC, and myeloid cells, pseudo-time trajectory analysis was performed with Monocle2[56–58]. The trajectory was visualized by plot_cell_trajectory. Top 30 highly variable genes of fibroblasts and mononuclear phagocytes were selected and clustered as pseudo-time-related DEGs.

**RNA velocity**. For RNA velocity, BAM file containing fibroblasts, SMCs, and mononuclear phagocytes and reference genome GRCh38 (hg38) {GRCm38 (mm10)} were used in the analysis with velocyto (v 0.2.3) and scVelo (v0.17.17) in python with default parameters. The result was projected to the UMAP plot from the Seurat clustering analysis for visualization consistency.

**Cell-cell interaction analysis (CellPhoneDB)**. Cell-cell interaction (CCI) was predicted based on known ligand–receptor pairs by Cellphone DB v2.1.0[59]. Permutation number for calculating the null distribution of average ligand-receptor pair expression in randomized cell identities was set to 1000. Individual ligand or

receptor expression was thresholded by a cutoff based on the average log gene expression distribution for all genes across each cell type. Predicted interaction pairs with $p$-value < 0.05 and of average log expression >0.1 were considered significant.

**Transcription factor regulatory network analysis (SCENIC).** Transcription factor network was constructed by single-cell regulatory network inference and clustering (SCENIC) R toolkit (ref) using scRNA expression matrix and transcription factors in AnimalTFDB[60]. The GENIE3 package predicted a regulatory network based on the co-expression of regulators and targets. RcisTarget package searched for transcription factor binding motifs in the given data. Genes involved in the predicted regulatory network were defined as a gene set, and AUCell package calculated the AUC value of the gene set to assess the activity of the regulatory network in cells.

**Differential proportion analysis.** Differential proportion analysis was performed based on the ratio change. Firstly, we got the proportion of each cell type or subtype by dividing the number of cells by the total number of cells. Then, the Log2-fold change was calculated between POP and control samples and |Log2-fold change| > 0.5 was considered as the threshold for significant change.

**Statistics and reproducibility.** We performed scRNA sequencing of the USL in three patients with POP and one control patient who had undergone a hysterectomy for hysteromyoma.

**Reporting summary.** Further information on research design is available in the Nature Portfolio Reporting Summary linked to this article.

## Data availability

Our scRNA-seq data are accessible in Gene Expression Omnibus under the accession number GSE250414. Source data underlying figures are provided in Supplementary Data 1 and 2. Other data are available from the corresponding authors on reasonable request.

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

## Acknowledgements

We are grateful to Singleron team for their continuous support. We would like to thank Editage (www.editage.cn) for English language editing. We are grateful to the financial support from The National Natural Science Foundation of China (Grant number 81971365), the special fund for Science and Technology Innovation Teams of Shanxi Province (Grant number 202204051002031), Fund Program for the Scientific Activities of Selected Returned Overseas Professionals in Shanxi Province (Grant number 20200007) and Research Project Supported by Shanxi Scholarship Council of China (Grant number HGKY2019095).

## Author contributions

M.S. designed and performed experiments, analyzed the data and wrote the paper. J.Z. designed and analyzed the paper. L.W. and Xiling L. analyzed the data. Xiaochun L., W.W., and Q.H. provided human samples. Xiaochun L. initiated and designed the study. All authors contributed to the article and approved the submitted version.

## Competing interests

The authors declare no competing interests.
