## [Peer Review File · Communications Biology]

Reviewers' comments:

Reviewer #1 (Remarks to the Author):

In this study, the authors aim to explore gene expression and cellular heterogeneity in the uterosacral ligament in POP. They constructed transcriptional profile of the uterosacral ligament and identified 10 major cell types using single-cell RNA sequencing. Furthermore, the authors performed subpopulation analysis of POP primary cells, explored differentially expressed genes, and performed pseudo-time and transcription factor analyses. They also dissected pattern changes of cell-cell interaction between stromal and immune cells. The authors thought this study provides insight into the molecular mechanisms of POP and valuable information for future research directions.

Concerns:

1. Concern about the reliability of the results. The authors only used one control patient to carry out this study. Considering the heterogeneity, I think the sample size is not enough for the cell ratio, DEG and cell-cell interaction analysis between POP and control group. Meanwhile, the authors need to show that whether the heterogeneity for cell composition and gene expression were present among 3 patients.
2. In Fig1, the legend for D and E is swapped.
3. In all figures, if the authors aim to show the gene expression, the gene name should be italicized. Meanwhile, most of gene names need to be italicized in the manuscript.
4. In Fig2H "the heatmap showing the differentially expressed genes of four smooth muscle cells subclusters", it did not show the differentially expressed genes in SMC_2 well.
5. In all the pseudo-time analysis, the authors need to perform RNA velocity analysis to define the trajectory direction.
6. In Fig2K and 2M, only dozens of differential expressed genes were shown. How to define the differential expressed genes along the trajectory. The authors need to clarify them in the methods or results part.
7. In Fig2 and 3, the functional pathways for each subgroups need to be shown in main figures.
8. In Fig3, the heatmaps did not support the subgroup results for the differentially expressed genes in some subgroups is not obvious.
9. In Fig3C, the trajectory contains too many branches, which is inconsistent with previous studies. The authors need to redo this analysis.
10. In Fig3F, the Jaccard Similarity is too low.
11. In Fig4B, the authors need to show the numbers of cell-cell communications in POP and control samples respectively using Cellchat to depict the decreased numbers
12. In Fig4C-F, the authors need to label the corresponding functions for the ligand-receptor pairs. In addition, the authors can show the ligand-receptor pairs in both POP and control samples in Fig4D-E.
13. In Fig5B and 5D, the authors need to show the p-values for the GO enrichment results.
14. The authors need to perform some experiments to verify the TFs or ligand-receptor pairs expression change between POP and control samples. In addition, for the authors define several subgroups in fibroblasts, SMCs, ECs and myeloid cells, the authors need to verify these subgroups.
15. The authors need to check the manuscript carefully to make sure the description is correct. For example, in Fig2 legend, "Subclustering of fibroblasts, SMCs and DCs in USL reveals cellular heterogeneity", DCs need to be revised to ECs.

Reviewer #2 (Remarks to the Author):

I want to thank the authors for this very interesting manuscript. They performed single-cell RNAseq analysis of cells isolated from the uterosacral ligament in patients with POP and control cells (patients who underwent hysterectomy for hysteromyomas). The authors identified 10 major cell types and have performed genetic expression and transcription factor analyses of these cell types.

Overall, the study was well conducted and, for the most part, well reported. I see two main limitations

of this work. One is the small sample size. Compared to other compatible studies (e.g. Li et al Nature Communications 2021), the authors have a much smaller sample size which limits generalisability and reproducibility of this work. The authors only have 3 POP samples and 1 control sample (in contrast to 16 POP and 5 control in Li paper) – the authors should justify the importance of this work despite this limitation. The other limitation is the authors have not highlighted the importance of the USL in the pathogenesis of POP. Without this, it is difficult to assess whether their study is a valuable contribution to the field.

Please find other specific comments below:

Abstract: The authors should revisit this abstract and provide some specific findings and implications of those findings for the pathogenesis of POP. Also mention how big the transcriptomic atlas is
Line 11: What are new clusters of uterosacral ligaments? Please re-phrase

Line 47: This sentence makes it sound like Li et al didn't look at the gene expression changes from tissue isolated from POP patients (they did). Instead, the authors should highlight that looking at the USL is what makes this study different.

Line 350-351: which previous studies have found that the main component of the USL is the ECM. Please provide citations to support this.

Line 416: Can the authors comment on the implication of their small sample size (3 POP and 1 control patient).

Reviewer #3 (Remarks to the Author):

Hence there was no study has analyzed the gene expression changes of the USL in POP by performing scRNA-seq thus far. This study is meaningful and could provide insight to improve prevention and treatment strategies of POP. However, there are still some problems to be solved.

1. There was a cellular heterogeneity between POP1, POP2 and POP3, please discuss it.
2. The subpopulation differences between the POP group and the control group were not analyzed. For example, which type of fibroblast was predominant in POP group.
3. Why Cell-cell interactions in the POP group were attenuated compared with those in the control group, please discuss it.

Response to Reviewer 1: Thank you for your review of our paper. We have answered each of your points below.

1. Concern about the reliability of the results. The authors only used one control patient to carry out this study. Considering the heterogeneity, I think the sample size is not enough for the cell ratio, DEG and cell-cell interaction analysis between POP and control group. Meanwhile, the authors need to show that whether the heterogeneity for cell composition and gene expression were present among 3 patients.

Response: As mentioned in our email dated 19 April, USL samples were obtained from resected uterine samples. However, the USL attached to the resected uterine specimen was a very small portion. In addition, the USL is rich in the extracellular matrix, and it is difficult to obtain qualified samples because of the high rate of cell clumping and the high percentage of impurities after the dissociation of USL samples. At present, we have not obtained additional qualified samples of USL for the control group, but we will not stop trying to obtain qualified samples.

As for heterogeneity for cell composition and gene expression between samples, we conducted Correlation analysis and showed as follows. However, it was found that there was no significant correlation between POP samples in cell composition and gene expression. Perhaps the occurrence of POP is only due to some specific changes, which can not be seen in the overall expression profile and cell composition.

Correlation analysis of cell composition

Correlation analysis of gene expression

In addition, we supplemented the clinical characteristics of patients (supplementary data 1), and our patients strictly adhered to the inclusion criteria and exclusion criteria.

2. In Fig1, the legend for D and E is swapped.

Response: Sorry for that. I have re-examined the content of the manuscript to make sure it is correct (Line 92-93).

3. In all figures, if the authors aim to show the gene expression, the gene name should be italicized. Meanwhile, most of gene names need to be italicized in the manuscript.

Response: As required, all gene names has been italicized.

4. In Fig2H “the heatmap showing the differentially expressed genes of four smooth muscle cells subclusters”, it did not show the differentially expressed genes in SMC_2 well.

Response: The legend of heatmap for SMCs, macrophages, monocytes and T cells was changed (Supplementary Fig. 4B-D, Supplementary Fig. 6B).

8. In Fig3, the heatmaps did not support the subgroup results for the differentially expressed genes in some subgroups is not obvious.

Response: Same as question 5, we adjusted the means of expressions that are not enough for clustering (Supplementary Fig. 4B-D, Supplementary Fig. 6B).

5. In all the pseudo-time analysis, the authors need to perform RNA velocity analysis to define the trajectory direction.

Response: As required, RNA velocity analysis has been performed and showed in our manuscript (Line125-128, 156-159, 234-234).

6. In Fig2K and 2M, only dozens of differential expressed genes were shown. How to define the differential expressed genes along the trajectory. The authors need to clarify them in the methods or results part.

Response: Top30 highly variable genes were selected as pseudo time-related DEGs and clustered. And we added this to the methods section (Line525-526).

7. In Fig2 and 3, the functional pathways for each subgroups need to be shown in main figures.

Response: We adjusted the figure to show the functional pathways for each subgroups (Figure 2D, Figure 3D, Figure 3J, Figure 4D, Figure 4H).

9. In Fig3C, the trajectory contains too many branches, which is inconsistent with previous studies. The authors need to redo this analysis.

Response: Based on the trajectory analysis of 3 subgroups or 5 subgroups of MP cells, we can get the trajectory with 3 branches, while based on the analysis of 11 subgroups, we can get the trajectory with 2 branches. So we finally presented the trajectory analysis results in 11 subgroups. The previous presentation may not be clear enough, but now we will show each cell group separately, hoping to be more intuitive.

10. In Fig3F, the Jaccard Similarity is too low.

Response: According the DEGs provided by Xie et al., we reconducted correlation analysis and gene set analysis (Figure 3F), and found that Neutrophils1 were similar with G3-5a, Neutrophils2 were in accord with G5c, and neutrophils3 were in accord with G5b subpopulations.

11. In Fig4B, the authors need to show the numbers of cell-cell communications in POP and control samples respectively using Cellchat to depict the decreased numbers

Response: We conducted cell-cell communications analysis using Cellchat, and showed as follow. However, there is no significant difference between Cellphone and Cellchat analysis results. The numbers of cell-cell communications in POP and control samples respectively using Cellphone were showed as follow.

Cellchat:

Control

POP

Cellphone:

Control

POP

12. In Fig4C-F, the authors need to label the corresponding functions for the ligand–receptor pairs. In addition, the authors can show the ligand–receptor pairs in both POP and control samples in Fig4D-E.

Response: We can't label the function of receptor ligand pairs in the diagram, so we add the explanation of their function in our manuscript. In addition, we added the comparison between POP and control groups in Cell-Cell interaction part (Line294-309).

13. In Fig5B and 5D, the authors need to show the p-values for the GO enrichment results.

Response: We revised GO enrichment analysis results with p values (Figure 7).

14. The authors need to perform some experiments to verify the TFs or ligand–receptor pairs expression change between POP and control samples. In addition, for the authors define

several subgroups in fibroblasts, SMCs, ECs and myeloid cells, the authors need to verify these subgroups.

Response: As mentioned in response to concern1, qualified USL samples were hard to obtain. When we got the USL sample, we will first consider dissociation to see if it meets the requirements of single-cell sequencing, so no further basic experiments can be carried out.

15. The authors need to check the manuscript carefully to make sure the description is correct. For example, in Fig2 legend, “Subclustering of fibroblasts, SMCs and DCs in USL reveals cellular heterogeneity”, DCs need to be revised to ECs.

Response: Sorry, I don't see error in the legend you mentioned. The original version is ECs. However, I have re-examined the content of the manuscript to make sure it is correct.

Response to Reviewer 2: Thank you for your comments. Our answers to your points are as follows.

1. Abstract: The authors should revisit this abstract and provide some specific findings and implications of those findings for the pathogenesis of POP. Also mention how big the transcriptomic atlas is.

Response: We revised the abstract to mention potential mechanisms and the size of the transcriptional map (Line7-17).

2. Line 11: What are new clusters of uterosacral ligaments? Please re-phrase

Response: We had reworded the wording and had not yet been able to prove a new cell cluster.

3. Line 47: This sentence makes it sound like Li et al didn't look at the gene expression changes from tissue isolated from POP patients (they did). Instead, the authors should highlight that looking at the USL is what makes this study different.

Response: Sorry for our poor expression, I changed the sentence to "Although Li et al. have used scRNA-seq to construct a transcriptomic atlas of anterior vaginal prolapse in POP using the vaginal wall and contributed to defining the molecular mechanism of POP, no study has analyzed the USL in POP by performing scRNA-seq thus far." (Line47-50).

4. Line 350-351: which previous studies have found that the main component of the USL is the ECM. Please provide citations to support this.

Response: We provided citations (Line376).

5. Line 416: Can the authors comment on the implication of their small sample size (3 POP and 1 control patient).

Response: As we mentioned in our previous reply, the USL is rich in the extracellular matrix, and it is difficult to obtain qualified samples because of the high rate of cell clumping and the high percentage of impurities after the dissociation of USL samples. After more than one year of our efforts, we finally obtained and successfully analyzed these 4 qualified samples. We will not stop trying to obtain qualified samples of the USL until the article is accepted, and even after it is accepted. We have taken two samples in these two months, but they were still unqualified after dissociation. Although we tried so hard to obtain samples, we still could not guarantee that we could obtain more qualified samples.

Response to Reviewer 3: Thank you for your comments. Our answers to your points are as follows.

1. There was a cellular heterogeneity between POP1, POP2 and POP3, please discuss it.

Response:

We have tried our best to control heterogeneity and followed the experimental process suggested by relevant methodology and literature. However, single-cell RNA sequencing is a technology with high technical difficulty, and factors such as sample source, RNA quality, and so on will affect the accuracy and consistency of the results. Therefore, in our research, we have taken many samples to improve the reliability of the data, and strictly controlled and screened the data. At the same time, we also used a variety of data analysis methods and algorithms to deal with heterogeneity to ensure the quality and stability of the obtained scRNA-seq data. We also described these steps and methods in detail in the article.

In addition, we believe that the existence of heterogeneity is a common problem in the current development stage of scRNA-seq technology, and it is reflected in many studies. Therefore, we will further verify them in various ways to ensure the reliability and wide applicability of the research results.

We further supplemented the clinical characteristics of patients (supplementary data 1), and our patients strictly adhered to the inclusion criteria and exclusion criteria. Thank you again for your comments on our research. We will seriously consider your feedback and make positive improvements.

2. The subpopulation differences between the POP group and the control group were not analyzed. For example, which type of fibroblast was predominant in POP group.

Response: We added the subpopulation differences in the SMCs, ECs, macrophages, and monocytes subpopulations, and added the corresponding figures (Line130-131, 175, 185-187, 200-201). As for fibroblasts, DCs, and other subpopulations, there were no significant difference or significance, so they were not explained in detail.

3. Why Cell-cell interactions in the POP group were attenuated compared with those in the control group, please discuss it.

Response: We conducted relevant discussions in the discussion section (Line426-429).

REVIEWERS' COMMENTS:

Reviewer #1 (Remarks to the Author):

The authors have addressed most of my concerns and I recommend its publication.

Reviewer #3 (Remarks to the Author):

The author has already revised as requested